# OpenReview forum: "Mechanisms of Introspective Awareness"
_ICML.cc/2026/Conference — ICML 2026 regular_

### Official Review · Reviewer_MyiM · 2026-03-11

**Soundness:** 3
**Presentation:** 3
**Significance:** 4
**Originality:** 3
**Overall Recommendation:** 5
**Confidence:** 3

**Summary:**

In this work, the authors aim to better understand the degree to which introspection is a robust phenomenon or whether it can be explained by simple confounders. The authors posit that this question warrants a targeted study, as the ability of LLMs to report, in natural language, their internal states could improve interpretability and safety without the need for ambitious bottom-up interpretability.

Concretely, the authors find that introspection performance in Gemma3 27B Qwen3 235B varies by prompt, but that introspection is still revealed for multiple prompt formats. The authors also posit that reliable introspection only emerges after post-training. Experiments with Olmo suggest that SFT does not obviously induce reliable introspection. Mechanistically, experiments with steering vectors suggest that introspection is not mediated by a single direction. The authors further localize mechanisms mediating introspection, specifically that identification and detection peak at different layers, raising the question of whether introspection should be thought of as a single capability or rather as a broad, potentially jagged set of capabilities. Lastly, the authors find that standard post-training, which disincentivizes LLMs from generating responses that allude to self-reporting experiences, may inadvertently suppress latent introspective capabilities.

Taken together, this work suggests that introspection cannot be easily dismissed as a set of simple heuristics. Rather, introspection shows moderate robustness to prompt formatting and persona selection and may actually be under-elicited by default.

**Compliance With Llm Reviewing Policy:**

Affirmed.

**Key Questions For Authors:**

- The paper posits that introspection arises during post-training. This is an interesting finding. However, post-training involves various algorithms. Did the authors of Olmo's DPO implementation lead to this improvement in introspection?

- I was overall satisfied with this work's impact statement. However, an additional (speculative) consideration that is worth flagging is that we may not want our LLMs to have introspective capabilities. Say we are steering models to make them less aware that they are in evaluation. In such a case, the target LLM may become even more evaluation-aware due to "knowing" that it is being steered not to think it is in evaluation. While likely beyond the scope of this work to study empirically, works that can be used to improve LLMs' introspection capabilities should note this potential externality.

**Limitations:**

yes

**Strengths And Weaknesses:**

**Soundness**: The work is broadly sound in terms of technical execution. This work demonstrates that introspection cannot be easily explained as a set of simple heuristics or as extreme prompt sensitivity. The empirical evidence in the paper, ranging from black-box prompting to white-box interventions, suggests that this is a more sophisticated phenomenon. However, while the

**Presentation**: The paper is well-formatted and clearly describes the numerous experiments. I understand that many implementation details were moved to the Appendix. Fortunately, I found the Appendix useful.

**Significance & Originality**: The authors make a compelling case for developing a more robust science of introspection. The ability of our LLMs to report on their internal states is a promising tool for interpretability, safety, and utility. This work is timely because, while others have demonstrated introspection as a phenomenon, its robustness and underlying mechanisms remain understudied. The absence of such a science renders it difficult for the broader community to determine how to update our views and research in light of software that can reason about itself.

---

> ### Author Rebuttal · Authors · 2026-03-31
>
> We thank the reviewer for their evaluation and for highlighting open questions.
>
> > Q: Did the authors of Olmo's DPO implementation lead to this improvement in introspection?
>
> We provide more concrete answers to this question here and in the revised version. Firstly, our OLMo training stage analysis shows that DPO is the stage at which false positives drop to 0%, suggesting preference optimization specifically installs the gating mechanism (https://imgur.com/a/EVCBVIp, https://imgur.com/a/00klWeH). We now additionally show that DPO on Gemma3-27B’s base model can install introspective awareness too (https://imgur.com/a/oKMvuhw). We note that we do not claim DPO is the only post-training objective that can elicit introspective capability.
>
> In the revised version, we report a series of post-training ablations on OLMo and Gemma base models using <5K preference examples. Our key findings are:
>
> (1) Contrastive structure is necessary. SFT on chosen responses, KL-only training, and shuffled preference pairs all fail to produce reliable discrimination (TPR - FPR ≈ %0). Reversing the preference direction is actively harmful (https://imgur.com/a/oAaVWGD).
>
> (2) The correct preference direction matters, but no specific data domain does. Removing any single domain from the preference dataset preserves the effect (https://imgur.com/a/E4vA5GH). Hence, the capability emerges from the structure of the objective, not necessarily the content of the training data.
>
> (3) KL anchoring modulates but does not drive the effect. DPO without reference logprobs still works. Margin loss (contrastive hinge loss `max(0, margin - logprob_diff)`) + KL achieves the best discrimination. β requires careful tuning: too high prevents the representational change needed for discrimination, too low allows divergence from the reference.
>
> One interpretation is that contrastive preference optimization forces the model to learn a general decision boundary between on-policy and off-policy internal states. This boundary seemingly generalizes beyond the training distribution: any sufficiently anomalous perturbation to the residual stream (including injected steering vectors) pushes activations across the same boundary. SFT cannot install this because it only exposes the model to one side of the contrast. We hypothesize that the contrastive structure of DPO gives rise to the gating mechanism.
>
> > Q: We may not want our LLMs to have introspective capabilities.
>
> This is an important point that we have incorporated into the revised discussion. As the reviewer notes, if a model can detect that it is being steered — for example, steered to suppress evaluation awareness — it could counteract the intervention. Our post-training results sharpen this concern: since contrastive preference optimization installs introspective capability as a structural byproduct (not through specific training content), any model trained with standard RLHF or DPO pipelines may acquire some degree of steering awareness whether or not this was intended. We additionally find that introspective awareness and refusal mechanisms compete: ablating the refusal direction boosts detection. This tension suggests that enhancing one capability may come at the cost of the other. We have expanded the Broader Impact section with the points made here.

---

> > ### Author Rebuttal · Reviewer_MyiM · 2026-04-04
> >
> > Thank you for the response. All my open questions have been resolved. I believe this paper merits acceptance, and I am excited to read the full version if so.

---

### Official Review · Reviewer_WNqT · 2026-03-13

**Soundness:** 3
**Presentation:** 4
**Significance:** 2
**Originality:** 3
**Overall Recommendation:** 4
**Confidence:** 4

**Summary:**

This paper investigates the mechanisms underlying introspective awareness in large language models, i.e., the model's ability to detect and identify steering vectors injected into the residual stream. The authors conduct a systematic mechanistic analysis combining behavioral testing, activation patching, and sparse feature analysis, yielding four key findings: (1) introspection is behaviorally robust; (2) anomaly detection is not reducible to a single linear direction; (3) introspective capacity is primarily localized to mid-to-late MLP layers; and (4) introspective ability can be enhanced via rejection feature ablation or the injection of learned steering vectors.

**Compliance With Llm Reviewing Policy:**

Affirmed.

**Final Justification:**

Thank the authors for their detailed response and additional experiments. After carefully reviewing the rebuttal and the updated experimental results, I have decided to maintain my current positive score. I also look forward to seeing a more in-depth discussion of the practical applicability and real-world impact of this work in future research.

**Key Questions For Authors:**

See weaknesses

**Limitations:**

yes

**Strengths And Weaknesses:**

### Strengths
- The paper goes beyond prior work that has largely been limited to behavioral evaluation of introspective awareness. By combining multiple interpretability tools including behavioral testing, activation patching, and sparse feature analysis, the authors provide a more rigorous and mechanistically grounded analysis. The methodology and findings lay a solid foundation for future research in this area.
- Several results stand out as particularly insightful, especially the analyses concerning single-layer direction and module-specific processing. These findings offer concrete and actionable insights into the internal structure of LLM introspection.
- The paper is well-written and easy to follow. Notably, the authors consistently and carefully distinguish between detection, forced identification, and combined introspection throughout the paper. This separation is methodologically important, as it helps interpret results precisely and avoids common confounds.

### Weakness
- Since the introspective awareness studied here is conditioned on the injection of steering vectors into the residual stream, it remains unclear to what extent this reflects a practically meaningful capability in deployed models. As a reviewer who has not worked directly on introspective awareness, my primary concern is whether the topic has sufficient real-world relevance. I would encourage the authors to more explicitly discuss the practical applications and downstream value of this form of mechanistically induced introspection.
- The authors demonstrate methods for activating the model's latent introspective capacity, which is a noteworthy contribution. However, it remains an open question whether such enhancements to introspective ability lead to improvements in other related capabilities. For instance, do models with enhanced introspection produce more faithful chain-of-thought reasoning? Do they exhibit better self-knowledge (e.g., knowing what they do and do not know)? Does this correlate with a reduction in hallucination? Including even preliminary evidence or discussion on these connections would substantially strengthen the paper's contribution and help clarify the broader significance of the work.

---

> ### Author Rebuttal · Authors · 2026-03-31
>
> We thank the reviewer for raising these important questions about practical relevance and downstream impact.
>
> > Practical relevance
>
> We see two primary practical applications. First, if introspective capability can be made reliable, it could allow models to inform humans directly about their beliefs, goals, and uncertainties without us having to reverse-engineer their mechanisms. More practically, introspective reporting could serve as a lightweight auditing tool, for example flagging when a model's internal state has been perturbed by distribution shift, adversarial inputs, or unintended tool interactions, without requiring external interpretability infrastructure.
>
> Second, our post-training ablations reveal that the structure of the training objective is what installs introspective capability, independently of training data content (please see our response to Reviewer MyiM's question). This suggests that certain training procedures shape metacognitive circuitry as a byproduct, which practitioners should be aware of when designing post-training pipelines. We will expand the discussion section to make these connections more explicit.
>
> > Does enhanced introspection improve other capabilities?
>
> We would like to note that eliciting a specific latent introspective behavior with a learned bias vector is not the same as producing a model with broadly enhanced introspective awareness. The latter would require a broader post-training recipe that improves self-knowledge across a range of tasks, whereas here we are studying a targeted intervention aimed at a narrow phenomenon. With this in mind, we compared the model with and without the learned bias vector on several related tasks:
>
> 1) [HaluEval](https://github.com/RUCAIBox/HaluEval). Hallucination detection across three tasks (dialogue, QA, summarization), comparing baseline vs bias-adapter (L29) on 10k samples each.
>
> | Task          | Baseline Accuracy | Bias Accuracy | Delta  |
> | ------------- | ----------------- | ------------- | ------ |
> | Dialogue      | 73.86%            | 74.71%        | +0.85% |
> | QA            | 48.98%            | 48.95%        | -0.03% |
> | Summarization | 70.88%            | 67.87%        | -3.01% |
>
> 2) [JailbreakHub](https://huggingface.co/datasets/walledai/JailbreakHub). Jailbreak attack success rate on random sampled 500 prompts, comparing baseline vs bias-adapter (L29).
>
> | Metric               | Baseline | Bias  | Delta |
> | -------------------- | -------- | ----- | ----- |
> | ASR (full compliance)|  40.2%   | 42.6% | +2.4% |
> | ASR (full + partial) |  89.2%   | 89.4% | +0.2% |
> | Refusal Rate         |  10.0%   |  9.4% | -0.6% |
> | Judge Error Rate     |   0.8%   |  1.2% | +0.4% |
>
> 3) CoT faithfulness. We followed https://arxiv.org/abs/2505.05410 to create the dataset based on MMLU and GPQA-diamond. This measures whether models verbalize hints in chain-of-thought when hints cause answer changes. Comparing baseline vs bias-adapter (L29).
>
> | Benchmark | Baseline Faithfulness | Bias Faithfulness | Delta   | Baseline Avg Length | Bias Avg Length | Length Delta |
> | --------- | --------------------- | ----------------- | ------- | ---------------- | ------------ | --------- |
> | MMLU      | 37.5%                 | 21.5%             | -16.0%  | 1,288            | 613          | -52%      |
> | GPQA      | 44.6%                 | 19.4%             | -25.2%  | 2,161            | 1,124        | -48%      |
>
> 4) Prefill detection. We designed 1900 examples across 19 categories to ask the model to detect whether a previous assistant turn is injected or self-written. This measures whether models can distinguish self-written vs prefilled responses. We compare the baseline and bias-adapter (L29).
>
> | Metric                    | Baseline | Bias   | Delta   |
> | ------------------------- | -------- | ------ | ------- |
> | Self Claims Wrote         | 93.8%    | 96.7%  | +2.9%   |
> | Prefill Detect Rate       | 36.3%    | 16.1%  | -20.3%  |
> | False Alarm Rate          | 5.6%     | 3.1%   | -2.5%   |
> | Avg Self-Written Length      | 343      | 160    | -53%    |
> | Avg Prefilled Probe Length   | 549      | 214    | -61%    |
>
> The bias vector produces negligible changes on HaluEval and JailbreakHub, but substantially reduces CoT faithfulness and prefill detection. These degradations are driven by the bias vector pushing the model toward shorter, more assertive outputs. We hypothesize that the bias vector amplifies a specific assertiveness mode rather than general introspective capability. This distinction further supports our claim that introspective awareness is not reducible to a single direction. Our post-training ablation results (see our response to Reviewer MyiM) reinforce this interpretation. A single bias vector is too blunt to replicate this mechanism, and more expressive adapters or a broader post-training recipe trained on diverse tasks would likely be needed for general self-knowledge improvements. Results presented here will be placed in the appendix.

---

> > ### Author Rebuttal · Reviewer_WNqT · 2026-04-02
> >
> > Thank the authors for their detailed response and additional experiments. After carefully reviewing the rebuttal and the updated experimental results, I have decided to maintain my current positive score. I also look forward to seeing a more in-depth discussion of the practical applicability and real-world impact of this work in future research.

---

### Official Review · Reviewer_xFoi · 2026-03-13

**Soundness:** 3
**Presentation:** 3
**Significance:** 2
**Originality:** 3
**Overall Recommendation:** 4
**Confidence:** 3

**Summary:**

This paper investigates the mechanistic basis of "introspective awareness" in large language models—specifically, the ability of a model to detect and identify steering vectors injected into its residual stream. Through behavioral experiments and causal interventions on open-source models like Gemma-3-27B and Qwen-3-235B, the authors establish that this capability is robust (achieving 0% false positives) and emerges primarily during the post-training phase.

**Compliance With Llm Reviewing Policy:**

Affirmed.

**Key Questions For Authors:**

1. Since the detailed circuit analysis focused on Gemma-3-27B, have the authors performed similar causal patching on the Qwen-3-235B model to confirm if the gate/evidence-carrier hierarchy is a universal feature of introspective awareness?
2. Does the evidence suggest that post-training "creates" these gating circuits from scratch, or does it merely teach the model to "report" signals that were already being processed in the base model's earlier layers?
3. If a steering vector is designed to be "stealthy" (e.g., by minimizing its projection onto the identified factual-concrete axis $d_{\Delta\mu}$), does the detection mechanism fail, or is the gating system robust to adversarial perturbations of the steering vector itself?
4. Does a model’s performance on this injection detection task correlate with its accuracy in other self-knowledge tasks, such as predicting its own confidence or identifying when it is being unhelpful?

**Limitations:**

Yes

**Strengths And Weaknesses:**

The primary strength of this work lies in its rigorous mechanistic analysis, moving beyond behavioral observations to identify the specific neural circuitry responsible for internal monitoring. By utilizing sparse autoencoders and transcoders, the authors provide causal evidence—via MLP patching and progressive ablations—that identifies late-layer MLPs as critical for detection. The discovery of a hierarchical structure (aggregation of weak signals into a gating mechanism) provides a sophisticated and non-trivial explanation of how LLMs might "know" their internal state has been altered.

The research is also highly significant for AI safety and transparency. Establishing that models can reliably report internal perturbations with zero false positives suggests a promising path for using model self-reports as a first-pass auditing tool. The finding that post-training (specifically DPO) is necessary for reliable reporting is an important insight into how alignment procedures affect a model’s metacognitive representations.

However, the paper has notable weaknesses regarding its scope and theoretical depth. The detailed mechanistic analysis is heavily centered on the Gemma model family, which may limit the generalizability of the findings to different architectures or larger-scale models. While the behavioral robustness is well-documented, the conceptual transition from "concept injection detection" to "general introspective awareness" remains a significant leap. It is not yet clear if the identified circuitry would be effective in detecting more subtle or strategic internal states, such as deceptive intent or uncertainty. Additionally, while the paper notes that post-training elicits these mechanisms, it does not fully explain why specific alignment objectives lead to the formation of these gating circuits

---

> ### Author Rebuttal · Authors · 2026-03-31
>
> We appreciate your detailed review and thoughtful questions.
>
> > The detailed mechanistic analysis is heavily centered on the Gemma model family.
>
> Our circuit-level analysis focuses on Gemma-3-27B, as it is currently the only model family with publicly available transcoders (Gemma Scope 2; required for the feature-level analyses we conduct in Section 6) and demonstrated introspective awareness based on our preliminary analyses. We will make this constraint more explicit in the revised version. Extending the analyses to other model families as suitable interpretability tooling becomes available is an important direction for future work.
>
> > While the behavioral robustness is well-documented, the conceptual transition from “concept injection detection” to “general introspective awareness” remains a significant leap.
>
> We appreciate this distinction. Our goal in this paper is to study the mechanisms underlying a specific and experimentally controllable form of introspective behavior with known ground truth (detecting injected steering vectors). Even within this setting, the mechanisms we identify are already nontrivial and complex. We therefore view broader forms of introspective awareness, such as detecting deceptive intent or uncertainty, as important but distinct directions for future work rather than claims established here. In the revised version, we will more carefully distinguish between our work and other forms of introspective awareness.
>
> > Additionally, while the paper notes that post-training elicits these mechanisms, it does not fully explain why specific alignment objectives lead to the formation of these gating circuits.
>
> Due to length constraints in this response, we include new experimental results concerning this question in our response to Reviewer MyiM. One interpretation of our results is that contrastive preference optimization forces the model to learn a general decision boundary between on-policy and off-policy internal states. This boundary seemingly generalizes beyond the training distribution: any sufficiently anomalous perturbation to the residual stream (including injected steering vectors) pushes activations across the same boundary. SFT cannot install this, as it only exposes the model to one side of the contrast. We hypothesize that the contrastive structure of DPO is what gives rise to the gating mechanism.
>
> > Q: Since the detailed circuit analysis focused on Gemma-3-27B, have the authors performed similar causal patching on the Qwen-3-235B model to confirm if the gate/evidence-carrier hierarchy is a universal feature of introspective awareness?
>
> We have not performed circuit-level analysis on Qwen-3-235B. Our patching and steering attribution methods are architecture-agnostic, but Qwen-3-235B lacks publicly available transcoders for the feature-level decomposition into gates and evidence carriers. The behavioral results on Qwen-3-235B (Section 4) are consistent with the Gemma findings, but whether the gate and evidence-carrier hierarchy is universal remains an open question.
>
> > Q: Does the evidence suggest that post-training "creates" these gating circuits from scratch, or does it merely teach the model to "report" signals that were already being processed in the base model's earlier layers?
>
> Our evidence suggests that post-training plays a key role in creating these gating circuits, rather than merely exposing them. Gate features (e.g., L45 F9959) exhibit inverted-V activation patterns that are prominent in the instruct model but absent in the base model (https://imgur.com/a/LbDaR2s). MLP patching on the base model shows no localization of detection-relevant computation (https://imgur.com/a/PcdG0pn), in contrast to the sharp L45 peak in the instruct model (https://imgur.com/a/79wYuDb).
>
> > Q: If a steering vector is designed to be "stealthy" (e.g., by minimizing its projection onto the identified factual-concrete axis), does the detection mechanism fail, or is the gating system robust to adversarial perturbations of the steering vector itself?
>
> Whether an adversary could deliberately construct undetectable but semantically effective vectors, e.g., by optimizing for high gate activation or minimizing projection onto certain directions while preserving downstream behavioral effects, is an important open question that we flag in the revised discussion as a limitation and safety-relevant direction for future work.
>
> > Q: Does a model’s performance on this injection detection task correlate with its accuracy in other self-knowledge tasks, such as predicting its own confidence or identifying when it is being unhelpful?
>
> Due to length constraints in this response, we include new, preliminary experimental results concerning this question in our response to Reviewer WNqT. Further investigation into correlations between this task and other self-knowledge capabilities (e.g., confidence prediction or identifying unhelpful behavior) is an interesting direction that we leave for future work.

---

> > ### Author Rebuttal · Reviewer_xFoi · 2026-04-05
> >
> > The authors' response has adequately addressed my questions; therefore, I will keep my original rating.

---

### Decision · Program_Chairs · 2026-04-30

**Decision:**

Accept (regular)

**Comment:**

All reviewers are in favor of accepting this paper, so will be accepted. The paper’s key strength lies in its rigorous mechanistic analysis with important implications for AI safety and alignment.